## RESEARCH ARTICLE

# Differential activity of transcription factors and neuronal effectors during the development of pikeperch brain

Radka Symonová[1,2,*], Tomáš Jůza[1], Million Tesfaye[1,3], Marek Brabec[4], Zuzana Sajdlová[1], Jakub Brabec[1] and Jan Kubečka[1]

**ABSTRACT**

Juvenile pikeperch (*Sander lucioperca*) undergo several ontogenetic shifts, the timing of which determines the survival of their first winter. The shift from planktivory to a more active piscivorous phenotype involves moving from pelagic to demersal habitat with more stimuli and hence potential brain functional reorganizations. During two consecutive years, we collected planktivores and piscivores with different body sizes between the years, recording distinct stages relative to the shift, and analyzed their whole-brain transcriptomes in an ecological context. We identified a distinct non-overlapping group of transcription factors (TFs) significantly upregulated in each phenotype: TFs upregulated in planktivores correspond to initial establishment of brain regions and overall architecture; TFs upregulated in piscivores correspond to the refinement of neurons and the formation of specific neuronal circuits. The planktivores independently of body size were characterized by interconnected activity of two TFs, fosab and junba. Gene set enrichment revealed extracellular matrix and collagen-related transcripts in piscivores from both years. A high activity of solute carrier (Slc) transporters was identified in the smaller-bodied piscivores. The neurotranscriptomics results reflected differences in body size and matched with ecological data and survival rates. The brain regulome indicated that body size differences translate into the specific gene activity of juvenile pikeperch.

**KEY WORDS: Growth phenotypes, Body size, Solute carrier transporters, *Klf* orthologs, Neurodevelopmental regulome**

## INTRODUCTION

Different embryos, even within a single clutch, develop at slightly different rates. Such asynchrony was documented in the development of zebrafish (*Danio rerio*) embryos fertilized simultaneously and incubated at optimal conditions (Kimmel et al., 1995). Asynchrony arises at the earliest stages, and as it becomes gradually more pronounced, particularly among individuals from different clutches, growth phenotypes can be distinguished (Goodrich and Clark, 2023). Similar asynchrony is known also in pikeperch [*Sander (Stizostedion) lucioperca* L.] from aquaculture in its early stages (Tönißen et al., 2024) and from free water (Jůza et al., 2014; Tesfaye et al., 2025).

Pikeperch (Percidae, Perciformes) is the main predatory large-body fish species of European eutrophic freshwater bodies and plays a key role by reducing planktivorous and omnivorous fish abundance (Dörner et at., 1999; Wysujack et al., 2022). While highly appreciated in aquaculture and angling fisheries in Central Europe (e.g. Policar et al., 2019; Colchen et al., 2020), pikeperch is an invasive and unwanted species in other parts of Europe (e.g. Balkan, Pavličević et al., 2016; Iberian peninsula, Ribeiro et al., 2021; UK, Nolan et al., 2024, Stakėnas et al., 2024). Pikeperch ontogeny, among others, involves a change in its food ecology during the first year of life – planktivorous fingerlings inhabiting pelagic undergo a profound ontogenetic shift towards piscivory and become demersal (Tesfaye et al., 2025). The ontogenetic dietary shift in fish is driven by reaching a body size that allows a fish to swallow the first fish prey (Sánchez-Hernández et al., 2019) and by availability of prey of suitable size (Ginter et al., 2011). Hence, body size is a key trait driving the fate of each individual fish (Keppeler et al., 2020). Reaching the body size that enables a fish to swallow the first fish prey is asynchronous in pikeperch and results in coexistence of at least two size classes (Tesfaye et al., 2025; Symonová et al., 2025). Decades of ichthyological surveys identified a complex body size-driven relationship between planktivores and piscivores – beside food ecology, they differ in their abundance, favored habitat and in the length of their existence. A certain fraction of planktivores switches to piscivory earlier or faster and then co-exists with their 'delayed' siblings (Jůza et al., 2014; Tesfaye et al., 2024, 2025, in press; Tesfaye, 2025). After the first winter, both planktivores and piscivores contribute to the new generation roughly equally despite huge differences in their abundance (the less numerous piscivores have higher chances of surviving the first winter than the more numerous but less developed planktivores, Tesfaye et al. (in press). This can be considered a maternally induced adaptive phenotypic response to unpredictable environments through increasing within-clutch offspring size variation, the bet-hedging strategy (Crean and Marshall, 2009). The development towards distinct growth phenotypes within a population is driven by gene activity of neurophysiological determinants during brain ontogeny including e.g. a metabolism-related individual's personality (Metcalfe et al., 2016), performance capacity of fish during stress (Madison et al., 2015) and neurohormonal appetite control (Deal and Volkoff, 2020).

To understand the neurodevelopmental drivers of this complex eco-morphological shift, we have initiated neurotranscriptomics research and identified Gene Ontology (GO) terms significantly upregulated in whole-brain transcriptomes of both phenotypes of juvenile pikeperch

[1]Institute of Hydrobiology, Biology Centre of the Czech Academy of Sciences, 370 05 České Budějovice, Czech Republic. [2]Faculty of Science, University of South Bohemia, Branišovská 1760, 370 05 České Budějovice, Czech Republic. [3]Faculty of Fisheries and Protection of Waters, South Bohemian Research Centre for Aquaculture and Biodiversity of Hydrocenoses, University of South Bohemia in České Budějovice, Zátiší 728/II, 389 25 Vodňany, Czech Republic. [4]Institute of Computer Science, Czech Academy of Sciences, Pod Vodárenskou věží 2, 180 00 Prague, Czech Republic.

*Author for correspondence (radka.symonova@gmail.com)

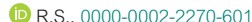 R.S., 0000-0002-2270-6018

(Symonová et al., 2025). However, neurotranscriptomics of juvenile non-model fish is only poorly understood. Hence, knowledge gaps and limitations in available resources persist despite the accumulating body of knowledge of its importance in mammals (e.g. Vinsland and Linnarsson, 2022). Only recently, a handful of studies have explored neurotranscriptomics of non-model juvenile fish species (Zhao et al., 2024 for common carp in response to warming; Ru et al., 2025 for greater amberjack under saline stress; Feng et al., 2025 in response to endocrine disruptors). Nonetheless, these studies were performed with fish from captivity that cannot reflect the gene activity of wild fish living under completely different conditions. Our study aims to fill this gap by analyzing wild fish.

Fish brain transcriptomes involve more than 20,000 genes (Symonová et al., 2025 for juvenile pikeperch); hence, it is useful to divide protein-coding genes into transcriptional regulators, i.e. transcription factors (TFs; Leung et al., 2022) and neuronal effector genes (NEGs, also known as neuronal terminal differentiation genes or neuron class effector genes; Zhang et al., 2021) encoding proteins with specific neuronal terminal features. TFs are key regulators of the NEGs governing the fate of diverse brain cell types; however, some TFs themselves can act as drivers of neuronal type. This underscores the multidimensionality of the vertebrate neural regulome and the need to distinguish further subcategories based on their known molecular functions during neurodevelopment: immediate early genes (IEGs) are critical components of brain development and function, playing key roles in neuronal activity, synaptic plasticity, learning, memory, and rapid responses to environmental stimuli (Tregub et al., 2025). IEGs in fish neurodevelopment include TFs (Calvo and Schluessel, 2021). In mammals, TFs became intensively studied (Wingender et al., 2018; Yin and Wang, 2014) and brain specific TFs are known mostly for model species (Sousa and Flames, 2022). Knowledge of fish brain specific TFs is still patchy and mostly related to food intake control in adult individuals of a handful of species kept in the laboratory (Vinnicombe and Volkoff, 2022). NEGs determine the functional properties of neurons and encode a wide array of proteins essential for neuronal functions: ion channels, G protein-coupled receptors, enzymes for neurotransmitter synthesis, synaptic proteins, transporters, receptors, neuropeptides and their receptors, cell adhesion molecules, motor proteins, and molecules of signal cascades (Hobert, 2011; Zhang et al., 2021). Of the NEGs, neuropeptides are chemical messengers involved among others in the appetite-controlling neuroendocrine systems in fish (Rønnestad et al., 2017; Deal and Volkoff, 2020). Neuropeptides with their receptors belong to transcripts upregulated in the juvenile pikeperch phenotypes (Symonová et al., 2025; this study). Composed of small chains of amino acids, these molecules are synthesized and subsequently released by neurons (Russo, 2017). Their primary

function lies in modulating neural activity, though their influence extends beyond the central nervous system (CNS). Neuropeptides require their specific receptors and exhibit distinct characteristics in their biosynthesis, release mechanisms, and the prolonged, often widespread, modulatory effects they exert in brain (Rønnestad et al., 2017; Deal and Volkoff, 2020). Anorexigenic and orexigenic neuropeptides significantly influence neurogenesis and neuronal migration, two fundamental processes in brain development; their relevance for pikeperch ontogeny is extensively discussed by Symonová et al. (2025).

Here, we first analyze newly obtained neurotranscriptomics data from two growth phenotypes of juvenile pikeperch of the season 2023. Then, we integrate data from both consecutive seasons 2022 and 2023 benefiting from non-overlapping body size ranges of the collected individuals. Results of ichthyological surveys provided a size-based frame related to their transition to piscivory and to their (quantitative) contribution to the next generation. Among the genes significantly differentially transcribed between planktivores and piscivores of both years, different groups of TFs and solute carrier transporters were prominent and complementary with two distinct neurodevelopmental programs determining the corresponding growth phenotype.

## RESULTS
### Ecological framework for neurotranscriptomics
We recorded two non-overlapping size stages of the planktivorous phenotype from the pelagic: larger planktivores closer to their transition to piscivory in 2022 and smaller planktivores farther from the transition in 2023 (Table 1, Fig. 1). Similarly, smaller demersal piscivores, i.e. sooner after their transition from planktivory, were recorded in 2023 and larger piscivores, i.e. later after their transition, were recorded in 2022. Planktivores were always much more abundant than piscivores (Fig. 1A). Their respective contribution to the 1-year-old cohort is evidenced in Fig. 1B – the larger and more numerous juvenile pikeperch of 2022 gave rise to an about three to four times stronger cohort of 1-year-old pikeperch in 2023 whereby the highest proportion was produced by the planktivores of August 2022 (likely continuation of cohorts is annotated by arrows at Fig. 1). The smaller and less numerous juvenile pikeperch of 2023 gave rise to a weaker cohort in 2024, where, however, original piscivores were more numerous than original planktivores.

The density of planktivorous pikeperch in late August 2022 was extremely high (682 individuals/hectare – Fig. 1A or 849 individuals/pelagic hectare). This high density, however, dropped many folds in just 3 weeks by 21 September 2022 (31 individuals/pelagic hectare) as recorded by our additional ichthyological survey in September 2022. This means that most pikeperch abandoned their pelagic way of life and either became demersal or died. Fig. 1B shows that an

**Table 1. Overview of individuals processed by whole-brain RNA-seq in years 2022 and 2023, and summary of upregulated genes obtained for phenotypes and years**

| Growth phenotype | Habitat | n | SL 2022 (mm) | SL 2023 (mm) | up 2022 | up 2023 |
|---|---|---|---|---|---|---|
| Planktivorous | Pelagic | 3+3 | 60-68 | 52-53 | 76 | 46 |
| Piscivorous | Littoral | 3+3 | 110-130 | 85-100 | 71 | 338 |

| Genes upregulated | Planktivores 2022 | Planktivores 2023 | Piscivores 2022 | Piscivores 2023 |
|---|---|---|---|---|
| Planktivores 2022 | × | 364 | 76* | × |
| Planktivores 2023 | 193 | × | × | 46 |
| Piscivores 2022 | 71* | × | × | 150 |
| Piscivores 2023 | × | 338 | 197 | × |

n, counts of individuals of the phenotype analyzed in 2022 and 2023; SL, standard length; up, significantly upregulated genes; counts of upregulated genes belong to the phenotypes in the left column. Lists of upregulated genes for each combination are in Tables S2-S4 and in Symonová et al. (2025). *Symonová et al. (2025).

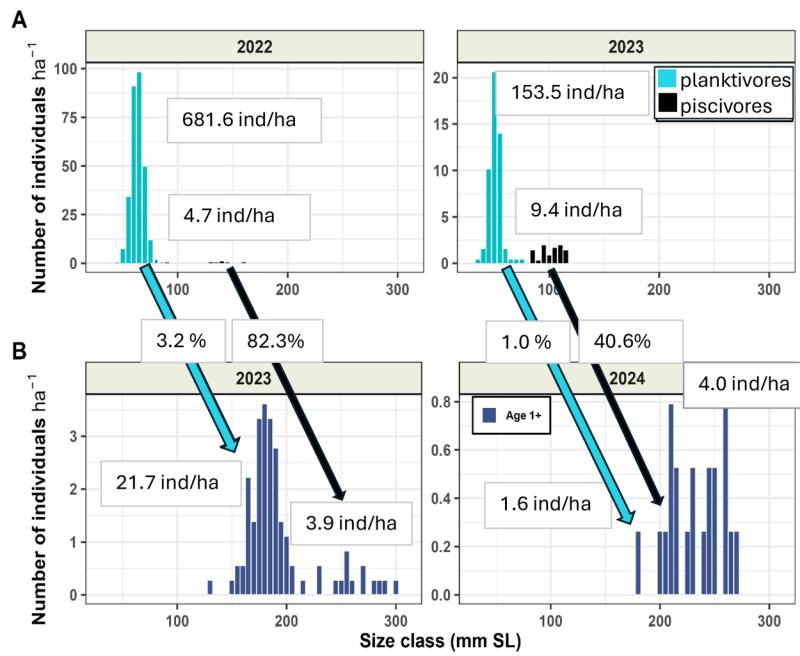

**Fig. 1. Overview of results of ichthyological surveys.** (A,B) Absolute densities of the planktivorous and piscivorous juveniles of pikeperch in size groups recorded during August surveys in the first (A) and in the second (B) year of life. Note different axes scales due to highly different densities of different groups in different years. Diagonal arrows represent the continuation of the same original phenotype in the 1st and 2nd year of life (in the 2nd year of life, all pikeperch are piscivorous). The proportions of second year pikeperch coming from planktivorous and piscivorous phenotype were calculated according to Tesfaye et al. (in press). Text boxes give overall densities of phenotypes in the reservoir and their percentual survival between the 1st and 2nd year of life. Planktivores are much more abundant than piscivores but the piscivores have much higher survival rates. ha, hectare.

important proportion (3.2%) of this extremely strong year class, i.e. individuals hatched in one year, in this case in 2022, survived till the next year. The ichthyological survey in 2023 confirmed that these individuals established the majority of the 2022-year class for its further development, where all individuals became piscivorous beginning with the next year of their life (Fig. 1B).

Below, we aim to characterize the differences between these two juvenile pikeperch phenotypes based on their whole-brain gene transcription and understand the potential reasons of their different success.

### Differentially transcribed genes between phenotypes and interannual comparison

There were 384 significantly differentially transcribed (DT) whole-brain genes between the two phenotypes in 2023 with 338 of them upregulated in piscivores (Tables 1, 2, Fig. 2A,B; Fig. S1A, Table S2). The higher count of DT genes in piscivores is reflected in the higher variance of their transcriptomes (Fig. 2C). Distances based on expression values show less similarities between phenotypes in each year (Fig. S1D,E) than within phenotypes between years (Fig. S1B,C,F-G). In 2022, 148 DT genes were recorded, with 72 of them in piscivores (details in Symonová et al., 2025). In 2023, the genes upregulated in piscivores were mostly related to collagen, its metabolism and the extracellular environment represented by GO terms cell periphery, extracellular matrix (ECM), extracellular region (ECR), and extracellular space (ECS; Fig. 3). The GO terms enriched among upregulated genes in piscivores are consistent between years (Symonová et al., 2025). The interannual comparison between piscivores in 2022 and in 2023 shows more similarities than between piscivores and planktivores in 2023 (Figs S1-S4). Different GO terms were enriched between piscivores 2022 and 2023 (Fig. S2) and planktivores 2022 and 2023 (Figs S3, S4). There is a reproducible and comparable pattern in the transcription rate between 2022 and 2023 genes and phenotypes (Tables S5, S6).

For piscivores of 2023, the STRING protein-protein interaction network analysis recognized 241 of the 274 uploaded and identified 241 nodes with 169 edges, an average node degree 1.4 and an

average local clustering coefficient 0.322 with an expected number of edges 65. The protein-protein interaction enrichment $P$-value was $<1.0e^{-16}$, and the network has significantly more interactions than expected. The GO terms enriched in this network in piscivores 2023 (Fig. 3) include Cellular components centered around ECM/ECR/ECS and collagen metabolism including four terms related to the membrane region. The Biological processes enriched are related to ECM, organogenesis, angiogenesis, cell differentiation and organismal development generally. The Molecular functions enriched are again centered around ECM and its activity, and transmembrane transport. The STRING Local Network Clusters enriched correspond to Mixed, incl. Telencephalon regionalization and Forebrain neuron differentiation (CL:26022), Mixed, incl. T-box TF-associated, special AT-rich sequence-binding, and ubiquitin-like domain superfamily (CL:26026), and Mixed, incl. Telencephalon regionalization and TF Orthodenticle homeobox protein 1 (CL: 26024). The Kyoto Encyclopedia of Genes and Genomes (KEGG) Pathways enriched were ECM-receptor interaction and Focal adhesion. The Reactome Pathways were ECM and Degradation of ECM, Slc-mediated transmembrane transport, Regulation of insulin-like growth factor (IGF) transport and Uptake by IGF-binding proteins, and Vascular Endothelial Growth Factor (VEGF) binding to

**Table 2. Summary of the differential transcriptomic analyses for both phenotypes and years**

| Transcript biotypes | 2022 | | 2023 | |
|---|---|---|---|---|
| | Planktivores | Piscivores | Planktivores | Piscivores |
| Total significantly upregulated transcripts | 76 | 72 | 46 | 338 |
| Protein coding genes | 66 | 71 | 36 | 306 |
| ncRNA | 3 | n/a | 1 | n/a |
| Uncharacterized protein coding | 2 | n/a | 4 | 23 |
| Uncharacterized ncRNA | 5 | 1 | 6 | 9 |
| Genes unknown for fish/ fish brain | 2+ | 4+ | 2 | 8 |

ncRNA, non-coding RNA.

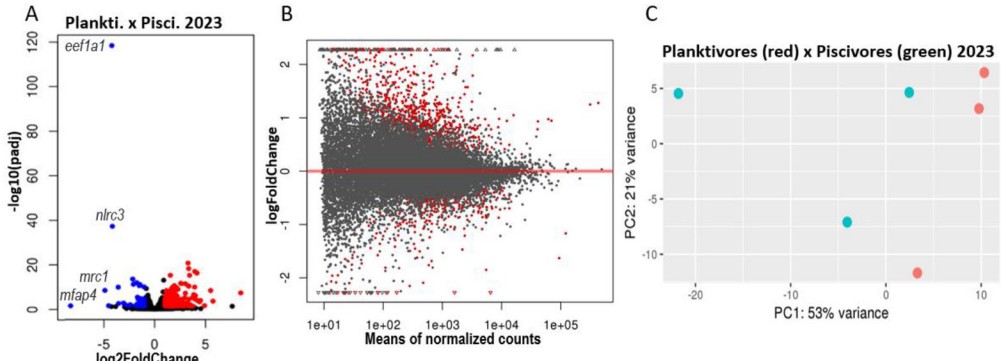

**Fig. 2. Overview of transcriptomic results between planktivores (*n*=3) and piscivores (*n*=3) in 2023.** (A) A volcano plot visualizing the global transcriptional change across the growth phenotypes. All transcribed genes are plotted; each data point represents a gene. The log2FoldChange of genes is on the *x*-axis and the −log10 of its adjusted *P*-value (padj) is on the *y*-axis. Genes with the padj value <0.05 and log2FoldChange >1 were upregulated in piscivores (red); genes with *P*adj <0.05 and log2FoldChange <−1 were upregulated in planktivores (blue). (B) The MA plot visualizes log2FoldChange in the mean expression between the phenotypes (letter 'M' in the plot name) compared to the mean value of normalized count of each gene (letter 'A' in the plot name). Genes with significantly differential expression (*P*<0.05) are highlighted in red, showing how expression levels change between two phenotypes (>0 upregulated in piscivores, <0 upregulated in planktivores). (C) The principal component analysis (PCA) plot depicts the transcriptomic similarity within and between the phenotypes; red dots are planktivores, green dots piscivores. PC1 and PC2 are principal components 1 and 2, respectively.

VEGF receptors leading to receptor dimerization. Interestingly, there is a similarity of the piscivores-upregulated brain transcripts to 'Fin morphogenesis' GO:0033334 GO Biological process, in which the anatomical structures of a fin are generated and organized.

A completely different situation exists in the planktivorous neurotranscriptomes from 2023. STRING recognized 28 transcripts of the 36 protein-coding genes, and their network of 28 nodes had merely three edges with average node degree 0.21 and average local clustering coefficient 0.107. Hence, the network does not have significantly more interactions than expected, and no GO term nor any other pathway was enriched among these significantly upregulated transcripts in planktivores.

The Cytoscape analysis of the season 2023 neurotranscriptomes identified the gene *faxdc2* (coding for a protein 'fatty acid hydroxylase domain containing 2') as the one with the highest Betweenness & Centrality value among the significantly upregulated transcripts in piscivores. This gene is, at the same time, unknown in fish brain, as listed below.

### TFs specific to planktivores and piscivores
Two non-overlapping sets of TFs were significantly upregulated between planktivores and piscivores in 2022-2023 (Table 3). The set of TFs upregulated in piscivores is dominated by homeobox genes (e.g. *arxa*, *emx1*, *emx3*, *lhx8*, *tbr1b*, *dlx3b*, *dlx5a*, *egr2*) with one basic helix-loop-helix (bHLH) factor (*bhlhe23*) and a distinct set of regulatory genes with general cellular roles related to growth, differentiation or tissue maintenance (*batf*, *aebp1a*, *macc1*, *osr1*, and *tbx18*). In planktivores, a distinct set of TFs with general cellular roles was upregulated (*batf*, *aebp1a*, *macc1*, *osr1*, and *tbx18*), mostly related to transcription elongation, protein modification, cell survival, and tissue remodeling. The *fosab* and *junba* TF transcripts were significantly upregulated (in 2022) and almost significantly upregulated (in 2023) in planktivores and belonged to the highly interconnected hub genes. They have several further paralogs transcribed in brain of both growth phenotypes in both seasons: *fosaa* and *fosl2* (the transcription rate of *fosb* and *fosl1a* is rather lower) and *jun*, *jund*, and *june*.

Interestingly, the TF annotated as Krueppel-like factor 15 (*klf15*) was present in both pikeperch stages in both years; however, it was represented by transcripts of three different genes residing in three

different genome loci and producing different proteins with different sizes and varying evolutionary conservation (Table 4). The two *klf15* transcripts upregulated in planktivores show more similarities to each other, since they are located in the same NC_050178.1 locus of chromosome 6 and in the same 5′→3′ direction (Table 4) although the genes are separated by several megabases. Further 14 genes of *klf* TFs were transcribed in 2023 and 12 in 2022 in both growth phenotypes, some of them with higher transcription rates equally in phenotypes and seasons (*klf7b*, *klf8*, *klf12b*, *klf13*).

The TFs with the highest transcription rate in both phenotypes and seasons are neuronal differentiation 1 (*neurod1*), followed by activating transcription factor 4a and 4b (*atf4a* and *atf4b*), Meis homeobox 2a (*meis2a*) and SRY-box transcription factor 11a (*sox11a*). Further TFs of the pikeperch neurotranscriptomes of 2022-2023, though not significantly differentially transcribed, produced in this study are in Table S5.

### Solute carrier transporters
Transcripts of solute carrier (Slc) transporters were another group of genes significantly upregulated in piscivores in 2023, whereby 20 members of 12 Slc families were identified in both years (Table 5). In planktivores, two Slc members of family 4 and 16, respectively, were significantly upregulated in 2023 (none of them in 2022). The two members belong to two families that were upregulated also in piscivores. Further Slc 334 genes of the pikeperch neurotranscriptomes of 2022-2023, though not significantly differentially transcribed, produced in this study are in Table S6. Of them, *slc1a2b* shows by far the highest transcription rate across years and phenotypes.

### Genes so far unknown to be transcriptionally active in fish and/or in brain
Transcripts of the following genes do not occur in the FishEnrichr database of transcripts so far known in fish (Kuleshov et al., 2016, 2019) nor in the Zfin database (Bradford et al., 2022) summarizing the genes currently known to be transcriptionally active in fish. They are thus for the first time reported to be transcribed in fish or fish brain.

Under gene symbol *faxdc2*, a transcript coding for fatty acid hydroxylase domain containing 2 was identified. This transcript was significantly upregulated in piscivores from 2023 and belonged to

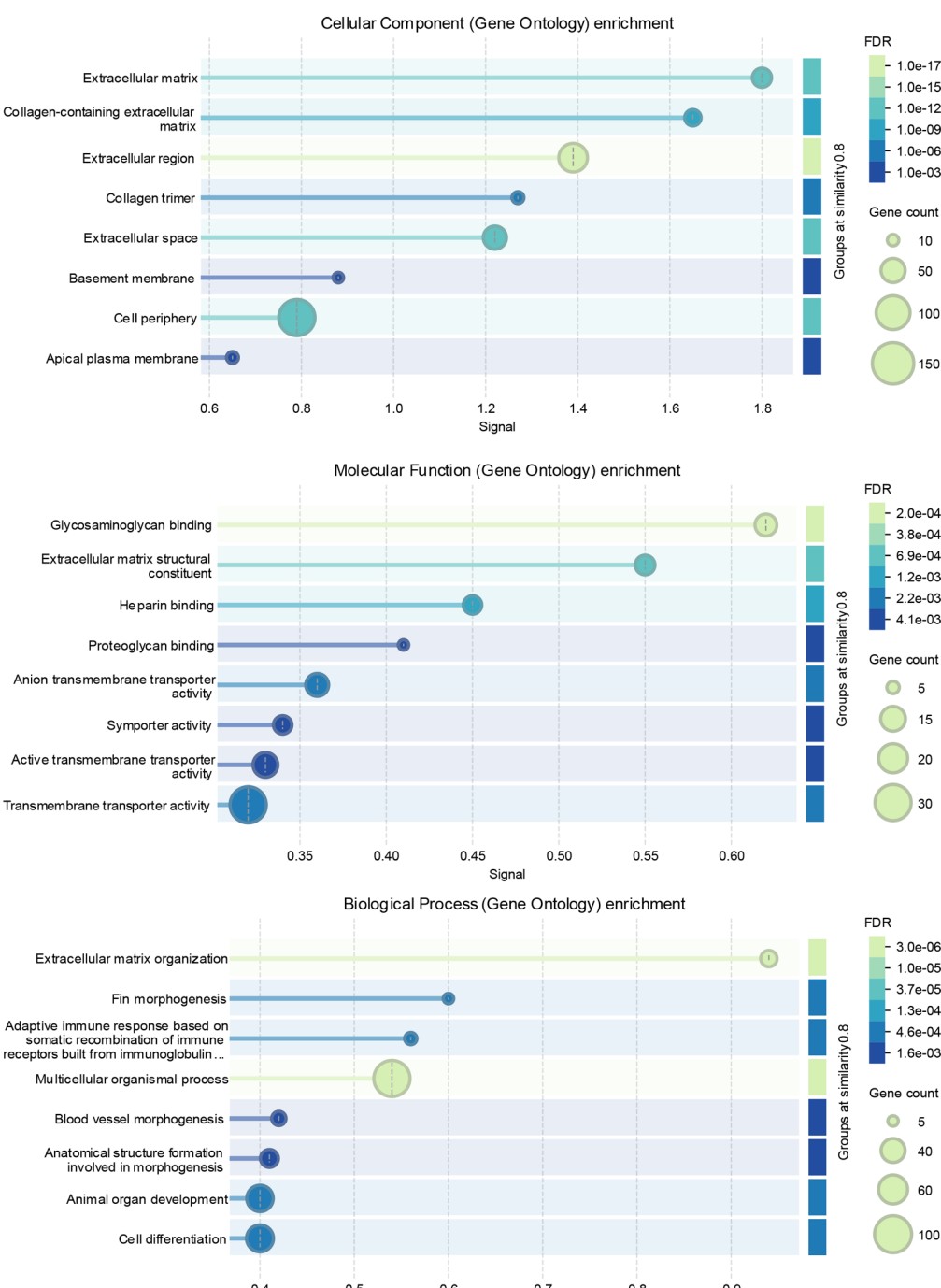

**Fig. 3. Intra-annual comparison of GO term enrichment between phenotypes in 2023.** Transcripts upregulated in piscivores of 2023, GO functional enrichment visualized by the STRING online tool (Szklarczyk et al., 2023). The eight most significant GO terms are shown for Cellular components, Molecular functions, and Biological processes. Terms grouped by similarity >=0.8, sorted by Signal. FDR, false discovery rate.

genes with the highest Betweenness & Centrality value. A kind of surprise was presence of the *pl* gene (LOC116056863) coding for protamine-like (Pl) protein. The gene was highly transcribed in both phenotypes and significantly upregulated in planktivores from 2023. A medium level of transcription was recorded in both phenotypes in 2022. The *krt13* gene (LOC116042622) coding for keratin, type I cytoskeletal 13-like showed a medium level of transcription and significant upregulation in piscivores from both years 2022 and 2023. The *ppk12* gene (LOC116054783) coding for mitogen-activated protein kinase 12-like showed a medium level of transcription. This transcript was significantly upregulated in piscivores in 2023 while comparably transcribed in both phenotypes in 2022. The *apoba* gene (*apob*, im:6911942, wu:fb30e06,

zmp:0000000614) coding for apolipoprotein Ba had a rather lower transcription activity. This transcript was significantly upregulated in piscivores in 2022 while comparably transcribed in both phenotypes in 2023. In parallel, this transcript showed high betweenness and centrality values in Cytoscape. The *matn2* gene (LOC116065768) coding for matrilin-2-like protein with a low level of transcription was significantly upregulated in piscivores in 2023 while missing in the dataset from 2022. The *cldn24* gene (LOC116059012) coding for a putative claudin-24 was upregulated in piscivores in 2023. The *cltrn* coding for collectrin, an amino acid transport regulator, was upregulated in piscivores in 2023. Similarly, the *vtxb* gene (LOC116037154) coding for a verrucotoxin subunit beta was upregulated in piscivores in 2023. Finally, the

**Table 3. Genes of TFs significantly upregulated in piscivores and planktivores in 2022-2023**

| Gene symbol | Synonym | Description |
| --- | --- | --- |
| **TFs upregulated in piscivores** | | |
| batf | si:ch211-153j24.4 | basic leucine zipper TF, ATF-like |
| alx4a | alx4a | ALX homeobox 4a |
| arxa | arxa | aristaless related homeobox a |
| bhlhe23 | bhlhe23 | basic helix-loop-helix family, member e23 |
| dlx3b | dlx3b | distal-less homeobox 3b |
| dlx5a | dlx5a | distal-less homeobox gene 5a |
| egr2 | LOC116056401 | E3 SUMO-protein ligase EGR2-like x early growth response 2a |
| emx1 | zgc:101821 | empty spiracles homeobox 1 |
| emx3 | emx3 | empty spiracles homeobox 3 |
| eomesa | tbr2, eom, eomes | eomesodermin homolog a |
| foxg1a | bf1 foxg1 brain factor 1 | forkhead box G1a, also known as fork-head TF family |
| klf11a | | Krueppel-like factor 11a |
| klf11b | | Krueppel-like factor 11b |
| klf15 | LOC116040748 | Krueppel-like factor 15 |
| lhx6b | si:ch211-236k19.2 | LIM homeobox 6b |
| lhx8 | LOC116063323 | LIM homeobox protein |
| macc1 | macc1 | MET transcriptional regulator (Metastasis Associated in Colon Cancer 1 MACC1) |
| osr1 | | odd-skipped related TF 1 |
| sp8a | cb459, fa07g05, sb:cb459 | sp8 TF a, a Znf TF |
| sp8b | Btd, buttonhead | sp8 TF b, a Znf TF of the Sp/KLF family |
| sp9 | sp9 | sp9 TF |
| tbx18 | tbx18 | T-box 18 TF |
| tbr1b | tbr1, zf-tbr1 | T-box brain TF 1b |
| glis1 | LOC116046141 | zinc finger protein glis1 |
| glis2 | LOC116042861 | zinc finger protein glis2-like |
| | LOC116052523 | zinc-binding protein A33-like |
| **TFs upregulated in planktivores** | | |
| cbx7a | | chromobox homolog 7a |
| fbxo33 | LOC116065701 | F-box protein 33 |
| fosab | c-fos, fos, cb1065 | v-fos FBJ murine osteosarcoma viral oncogene homolog Ab |
| gsx1 | gsx1 | Genetic Screen homeobox protein 1 |
| hlx1 | | H2.0-like homeobox 1 (*Drosophila*) |
| hoxc8a | | homeobox C8a |
| irx4a | ziro4a, zgc:77362 | iroquois homeobox 4a |
| irx5a/irx5b | LOC116055792 | iroquois homeobox 4a |
| junba | LOC116039550 | JunB proto-oncogene, AP-1 TF subunit a, TF jun-B-like |
| klf15 | LOC116050231 | Krueppel-like factor 15 |
| si:ch211-117k10.3 | klf15 | Krueppel-like factor 15, predicted to enable metal ion binding |
| nkx2.4a | fi48a12, wu:fi48a12 | NK-2 homeobox 4a |
| pitx2 | | paired-like homeodomain TF 2 or pituitary homeobox 2 |
| tefb | | transcription elongation factor TEF TF, PAR bZIP family member b |
| vdra | | vitamin D3 receptor A |
| vdrb | | vitamin D3 receptor B |
| | LOC116056321 | zinc finger protein 395-like |
| znf395b | | zinc finger protein 395b |
| znf704 | | zinc finger protein 704 |

*gyc88e* gene (LOC116049087) coding for a soluble guanylate cyclase 88E-like was highly transcribed and significantly upregulated in piscivores from 2023.

## DISCUSSION

The strikingly different counts of significantly DT genes between planktivores and piscivores in 2023 can be ascribed to the smaller body size of both phenotypes in 2023 when compared to 2022 (Table 1). Body size is one of the most important determinants for survival in the early stages, since a fish's survival depends on its ability to avoid predation and compete for food (Goodrich and Clark, 2023), and it varies considerably in juvenile pikeperch between years (Tesfaye, 2025). The smaller planktivores of 2023 were sampled during an earlier phase of a lower brain activity than the larger and obviously more advanced (larger) planktivores in 2022 (46 DT genes in 2023 versus 76 DT genes in 2022, Tables 1, 2, Fig. 4), whereas piscivores in 2023, although also smaller than in 2022, were sampled

during a phase of highly dynamic changes in gene transcription linked to their recent transition to piscivory, boosted by the higher food quality and more structured and challenging environment (338 DT genes in 2023 versus 71 DT genes in 2022, Tables 1, 2). The high differential gene activity of freshly switched piscivores of the 2023-year class indicates that the transition period may be an extremely important qualitative change of the whole brain transcriptome (Fig. 4). Nonetheless, it remains to be identified what specific brain region is responsible for the increase in gene transcription. The transition to piscivory during the summer (before our August sampling) is indeed advantageous as the life expectancy of August piscivores is about 25 times higher than that of August planktivores (Tesfaye, 2025). The switch to piscivory is thus an extremely important event from both ecology and transcriptomics point of view and deserves detailed attention in future studies.

No matter how advantageous the early transition to piscivory is, most of pikeperch in Lipno reservoir remain still pelagic and

**Table 4. Genes of the klf15 TFs with significantly different transcriptions between planktivores and piscivores**

| | | Planktivores 2023 | Planktivores 2022 | Piscivores 2023 |
|---|---|---|---|---|
| Gene/transcript | Annotation NCBI | *klf15* | *klf15/klf15l* | *klf15* |
| | Gene ID NCBI | LOC116050231 | si:ch211-117k10.3, 116036683 | LOC116040748 |
| | mRNA NCBI | XM_031300197.2 | XM_036002353.1 | XM_031286363.2 |
| | Locus NCBI | NC_050178.1 | NC_050178.1 | NC_050184 |
| | Span on the locus | 16,033 - 36,158 | 23,396,162 - 23,407,818 | 14,803,408 - 14,817,088 |
| | Direction | 5′→3′ | 5′→3′ | 3′→5′/complementary |
| | Size of the locus | 20,126 nt | 11,657 nt | 13,681 nt |
| | Chromosome | 6 | 6 | 12 |
| | Exon count | 3 | 3 | 3 |
| | Normalized gene hit counts (mean for three individuals) | 150 | 354 | 240 |
| Protein | Amino acids | 463 | 421 | 549 |
| | Protein ID NCBI | XP_031156057.1 | XP_035858246.1 | XP_031142223.1 |
| | Conserved domains according to Wang et al. (2023) | KLF15_N | KLF15_N | KLF15_N |
| | | COG5048 - FOG znf | COG5189/SFP1 – putative transcriptional repressor regulating G2/M transition | COG5189/SFP1 – putative transcriptional repressor regulating G2/M transition |
| | | 3× C2H2 znf | 3× C2H2 znf | 3× C2H2 znf |
| | | putative nucleic acid binding site | putative nucleic acid binding site | putative nucleic acid binding site |
| | | zf-H2C2_2 – znf double domain, pfam13465 | | |
| | Molecular weight | 50387.87 | 45270.79 | 58805.55 |
| | Theoretical pI | 6.26 | 5.78 | 5.77 |
| | Instability index (II) | 75.22 (unstable) | 74.41 (unstable) | 71.74 (unstable) |
| | Aliphatic index | 71.6 | 67.48 | 71.42 |
| | | | | −0.421 |

nt, nucleotide.

planktivorous in late August of the first year of life and face the challenge of transition to piscivory. In line with this, a great majority of the abundant 2022-year class was still composed of planktivores in August (Fig. 1A). However, their body size was sufficient to switch to piscivory in September. They had just a few weeks to accomplish the transition before the onset of the cold period of the year when plankton would be scarce. The September 2022 survey showed that planktivores vanished from pelagic, which is the first precondition of becoming piscivorous. The assessment of 2-year-old fish in 2023 (Fig. 1B) showed that a good proportion of planktivores successfully switched to piscivory and they made the base of a strong 2022-year class (Tesfaye, 2025).

In the long-term comparison of juvenile pikeperch body size, year 2023 with smaller individuals of both phenotypes was rather average and reflected the most usual situation, while year 2022 was an exception in terms of body size (Tesfaye et al., 2024;

**Table 5. Overview of the Slc genes significantly upregulated in piscivores and planktivores in 2022-2023**

| Up | Slc | Synonym (year) | Family function/description |
|---|---|---|---|
| piscivores | *slc13a3* | slc13a3 | Na⁺-sulfate/carboxylate cotransporter |
| | *slc13a4* | slc13a4 | Na⁺-sulfate/carboxylate cotransporter |
| | *slc16a12b* | LOC116061615 | monocarboxylate transporter | MCT |
| | *slc16a4* | slc16a4 | monocarboxylate transporter | MCT |
| | *slc1a1* | LOC116039652 | high-affinity glutamate and neutral AA transporter |
| | *slc1a6* | slc1a6 | high-affinity glutamate and neutral AA transporter |
| | *slc22a2* | LOC116060335 | organic cation/anion/zwitterion transporter |
| | *slc22a6* | LOC116034895 | organic cation/anion/zwitterion transporter |
| | *slc22a6* | LOC116062370 | organic cation/anion/zwitterion transporter |
| | *slc23a2* | si:dkey-106n21.1 | Na⁺-dependent ascorbic acid transporter |
| | *slc23a4* | zgc:110789, im:7144096 | Na⁺-dependent ascorbic acid transporter |
| | *slc25a4* | 2022 | mitochondrial carrier |
| | *slc2a11a* | slc2a11a | facilitative GLUT transporter |
| | *slc35f2* | slc35f2 | nucleotide-sugar transporter |
| | *slc39a8* | slc39a8 | metal ion transporter |
| | *slc47a1* | slc47a1 | multidrug and toxin extrusion |
| | *slc4a10b* | slc4a10b | bicarbonate transporter |
| | *slc4a2b* | slc4a2b | bicarbonate transporter |
| | *slc4a5b* | slc4a5b | bicarbonate transporter |
| | *slc5a5* | slc5a5 (2022-2023) | Na⁺- and Cl⁻-dependent Na:neurotransmitter symporters |
| planktivores | | | |
| | *slc16a1a* | si:ch211-105j21.7 | monocarboxylate transporter | MCT |
| | *slc4a1a* | zgc:111889, zgc:152771 | bicarbonate transporter |

AA, amino acid.

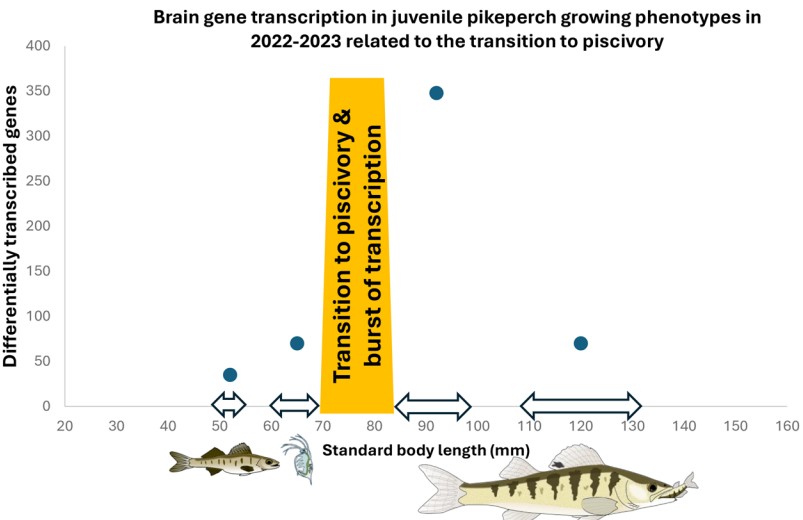

**Fig. 4. Brain gene transcription in pikeperch growing phenotypes in 2022-2023 related to the transition to piscivory.** Visualization of ecological data is integrated with neurotranscriptomics for both years and growth phenotypes.

Tesfaye, 2025). Despite the non-overlapping body size ranges, the GO terms enriched among the genes significantly upregulated in piscivores are consistent between the years. The comparable pattern in the transcription rate between 2022 and 2023 genes and phenotypes (e.g. Tables S5, S6, overall transcription of TFs and Slc genes) further supports the validity of our results despite a higher intra-phenotype variance. The upregulation of Slc genes in piscivores from 2023 can be ascribed to the abrupt change in food quality and quantity upon the recent transition to piscivory – activity of Slc genes is linked to and reflected by food intake as well as by increased energy demand (Nguyen et al., 2021) in newly piscivorous brain facing new challenges.

The body size of juvenile pikeperch determines whether an individual becomes able to catch its first fish prey or not. Hence, growth is regulated by intrinsic neuroendocrine mechanisms integrating signals from the gastrointestinal tract in brain (Soengas et al., 2018). Yet, growth also largely depends on extrinsic conditions that obviously influenced the 2023 pikeperch population and its overall success. As a result, the two growth phenotypes in both consecutive years were determined by their distinct whole-brain gene transcription in response to intrinsic and extrinsic factors.

### TFs in the juvenile pikeperch brain

TFs are essential among others for vertebrate neurodevelopment (Leung et al., 2022) and potential regulators of appetite in fish (Vinnicombe and Volkoff, 2022). The two non-overlapping sets of TFs significantly upregulated in planktivores and piscivores, respectively, represent a distinct group of homeobox, Klf and Znf TFs in each phenotype. TFs of the groups Batf, Bhlhe, Egr, Foxg, Glis, Macc, Osr, Sp, Tbx, and Tbr are upregulated in piscivores, and Fbxo, (proto)oncogene fosab and junba, Tef and vitamin D3 receptors are upregulated in planktivores. No TF transcript was upregulated in a year for planktivores and another year for piscivores and vice versa. Although the TF *klf15* was upregulated in both phenotypes, it was represented by three different paralogs. The transcription profiles of these TFs thus indicate their specific roles in neurodevelopment of the corresponding growth phenotype.

In planktivores, two (proto)oncogene TFs, *fosab* and *junba*, were highly transcribed in both seasons independently of the body size. They belong to the class leucine zipper factors (bZIP; Wingender et al., 2018) and were identified as the highly interconnected hub genes upregulated in the planktivorous phenotype in 2022 (Symonová et al., 2025) and in 2023. This is supported by the

STRING database providing further evidence of their protein-protein interactions (Szklarczyk et al., 2023). Moreover, these two TFs participate in the formation of another TF, the Activating Protein 1 (AP-1) complex that is a crucial marker of neuronal activation with numerous roles in neurodevelopment (Alberini, 2009). *Fosab* functions are linked to learning, memory, and the overall neuronal activity (Wang et al., 2025; Tregub et al., 2025; Kubra et al., 2022), which is relevant for the planktivorous pikeperch phenotype striving to catch its first fish prey. *Fosab* belonging to IEGs is rapidly activated in response to extracellular stimuli with key roles in neuronal activity and brain development in mammals as well as in fish (Calvo and Schluessel, 2021; Tregub et al., 2025). A pleiotropic role of *fosab* affecting both the brain physical architecture and its cognitive capabilities is known in zebrafish (Wang et al., 2025). *Junba* identified as a 'JunB proto-oncogene, AP-1 TF subunit a' also belongs to IEGs. Their bZIP domain facilitates their dimerization to form the AP-1 TF complex (Shaulian and Karin, 2002). The AP-1 is a pivotal regulator of gene expression and coordinates cell growth, proliferation, and differentiation by its involvement in both basal and inducible gene transcription (Alberini, 2009). In the CNS, AP-1 is essential for learning, memory, motor control, and cognition (Herdegen and Leah, 1998). FOS proteins can only heterodimerize with members of the Jun family, while Jun proteins can both homodimerize and heterodimerize with other Jun or Fos members to form functionally active complexes. Jun proteins also heterodimerize with other TFs such as members of the ATF family (Hai and Hartman, 2001), which have the highest transcription rate in the juvenile pikeperch brain (this study). Since interactions between *Fosab* and *Junba* are critical for brain development (West and Greenberg, 2011), their increased activity in planktivores may represent a distinct neurodevelopmental stage governed by these TFs regulating downstream regulatory and effector genes. For piscivores, another distinct group of TFs represent the major regulators, while Slc transporters are the major NEGs potentially regulated by TF and ensuring, among others, metabolic demands of the brain of the piscivorous phenotype.

The Krüppel-like factors (Klfs) are TFs with zinc finger (Znf) DNA binding domains known to play important roles in brain development and regeneration of CNS and optic nerve including adult zebrafish (Bhattarai et al., 2016; Veldman et al., 2007). Klf members are known to be developmentally regulated in the CNS and as transcription activators or repressors according to the specific cellular context. However, most studies on Klfs focused on non-nervous tissues of mammals (Moore and Goldberg, 2011;

Moore et al., 2011). Despite their ubiquitous transcription, not much is known about this TF family in the (juvenile) fish brain. *Klf6a* and *Klf7* exhibited wide and similar expression in the adult zebrafish CNS (Bhattarai et al., 2016). Recently, Chen et al. (2021) re-annotated 24 grass carp klf genes and analyzed them in the context of a virus infection in immune-relevant tissues. Of the *klf11* and *klf15* upregulated in juvenile pikeperch brain, *klf15* is an outlier that was not assigned to any specific Klf subfamily because of limited knowledge about its protein interaction motifs and largely unknown functions (Moore et al., 2011; Xiang et al., 2024). A study using bovine retinal cDNA library reports Klf15 acting as a transcriptional repressor of the rhodopsin and interphotoreceptor retinoid-binding protein promoters and being involved in restraining photoreceptor-specific gene expression in non-photoreceptor cells (Otteson et al., 2004; Xiang et al., 2024). Klf11 defined as a transforming growth factor (TGF) beta-inducible gene belongs to the SID-R2/3 subfamily of Klf (Moore et al., 2011). It is known to be involved in fat metabolism in mouse liver, glucose metabolism in pancreas, cardiac fibrosis and cancer, i.e. not in nervous tissue (Xiang et al., 2024; Yuce and Okzan, 2024). Since the three different *klf15* orthologs were upregulated between piscivores and planktivores, they may exert potentially important, however, currently unknown functions for the respective phenotypes. Thus, Klfs together with other TFs are a promising group of transcriptional regulators for downstream neurodevelopmental research in (juvenile) fish brain since they provide insight into the role of paralogs that emerged also in genes of Slc transporters. Paralogs originating from the teleost-specific genome duplication (Glasauer and Neuhauss, 2014) increase the overall count of transcribed genes and therefore potentially enrich the regulatory landscape.

### The extracellular space of the piscivorous pikeperch brain

The brain tissue is composed of two main components: cellular elements (neurons and glial cells) producing the gene transcripts, and the gap between the elements, the ECS crucial for the proper functioning of the elements. The most important terms of all three GO categories consistently upregulated in piscivores center around ECM, ECR, collagen-related terms and cell periphery, all closely linked to the ECS (Ashburner et al., 2000; The Gene Ontology Consortium, 2023; Symonová et al., 2025). ECM is one of fifteen 'External encapsulating structures' belonging to the Cellular component (CC) term 'Cellular anatomical entity: cell periphery'. With its synonym matrisome, ECM is defined as the structure lying externally to cells, which provides structural support and biochemical or biomechanical cues for cells. The cell periphery itself is a broad region including the cell membrane and any external encapsulating structures. ECR belongs to the CC term 'Cellular anatomical entity' and is defined as the space external to the outermost structure of a cell. This term annotates gene products secreted from a cell but retained within the organism (i.e. released into the interstitial fluid or blood). The ECS also known as intercellular space belongs to the CC term 'cellular anatomical entity'. ECS is the part outside the cells occupied by fluid. The width of the ECS is about 20-60 nm while it occupies ~20% of the brain volume (Kamali-Zare and Nicholson, 2013). The proximity to the cell membrane makes ECS and its content important for cellular homeostasis and function. The ECS contains a fluid similar to that found in brain ventricles that maintains an ionic balance across the cell membrane. The ionic balance establishes the cellular resting potential and permits neuronal action potential and synaptic transmission. The ECS provides a communication channel between cells for chemical signals (Nicholson and Hrabětová, 2017). The ECS accommodates an ECM forming a meshwork of long-chain polymeric molecules including chondroitin sulfate, heparin sulfate, etc. (Nicholson and Hrabětová, 2017). The transmembrane movement of diverse molecules is determined among others by size of the ECS and by the active transport that is regulated by Slc transporters. Transcripts of Slc transporters were significantly upregulated in the smaller-sized piscivores from 2023 in contrast to 2022 and to planktivores of both years. This indicates a more intense transmembrane movement in the 2023 piscivores and more interactions between brain cellular elements and their ECM. Hence, an overall higher brain activity in the smaller-sized 2023 piscivores can be inferred from their gene transcription that was then reflected in their higher survival in comparison to their underdeveloped planktivorous siblings (Fig. 1).

### Slc transporters in the piscivorous pikeperch brain

The Slc transporters are the largest family of transmembrane transporter proteins well characterized in mammals and particularly in humans, where over 400 members are organized into 66 families (Hediger et al., 2004; Perland and Fredriksson, 2017; Hu et al., 2020). Slc transporters facilitate the movement of diverse substrates across membranes encompassing ions, amino acids, nutrients, metabolites, nucleotides, and vitamins. In the brain, Slc transporters are fundamental for maintaining neuronal excitability, ensuring neurotransmitter homeostasis, supporting energy metabolism, and enabling detoxification processes. Their proper function is critical for brain development, cellular health, maintaining brain homeostasis, facilitating nutrient supply, enabling waste removal, regulation of neurotransmission, and overall neurological functions (Nguyen et al., 2021). Their significantly higher transcription in piscivores indicates a more dynamic transmembrane transport in this phenotype, i.e. a brain undergoing active maturation, characterized by high metabolic demand, ongoing neurogenesis, synaptic refinement, and precise regulation of ion and nutrient balance. Since blood vessels in the main parts of the CNS are impermeable to water-soluble molecules due to blood-brain barrier (BBB), specialized transport systems exist of which the protein-mediated transport is facilitated by Slc transporters among others. Slc transporters thus import nutrients and export metabolites through the BBB (Nguyen et al., 2021). The significant upregulation of Slc aligns with the importance of nutrients transport for sustaining brain function and supporting its growth in predatory piscivores. We provide first indications that the brain with upregulated transcripts of Slc genes is not merely growing but could be actively engaged in complex metabolic, homeostatic, and signaling processes that necessitate precise control over the intra- and extracellular environment. The variety of substrates handled by the upregulated Slc transporters also suggests an active and dynamic metabolic state which is expected in piscivores facing new ecological challenges. Indeed, the brain, especially during development, has substantial energy demands reflected among others by preference for littoral habitats (Závorka et al., 2023). We anticipate that developing brain needs to continuously import nutrients, efficiently remove waste products, maintain ion gradients crucial for neuronal excitability, and precisely regulate neurotransmitter levels. This transcription profile thus can be considered to match with ongoing processes such as neurogenesis, synaptogenesis, myelination, and the robust metabolic support required for these dynamic changes. However, these expectations need to be confirmed by functional experiments.

### Genes so far unknown to be active in fish and/or brain

Functions of the *Faxdc2* encoded fatty acid hydroxylase domain containing 2 are explored in mouse (The UniProt Consortium,

2017) and involve the following GO terms: CC - Membrane, MF - Iron ion binding and oxidoreductase activity, and BP - Sterol biosynthetic process. For zebrafish, the following terms are known CC - Endoplasmic reticulum membrane, MF - Methylsterol oxidase activity and iron ion binding, and BP - Sterol biosynthesis (The UniProt Consortium, 2017). According to FishEnrich, *faxdc2* is involved in BPs Membrane lipid biosynthesis and metabolism and Regulation of neurotransmitter secretion (Kuleshov et al., 2016, 2019). Regarding the highest betweenness and centrality values of this transcript and its involvement in sterol metabolism and iron ion binding, Faxdc2 should be further explored particularly in the piscivorous juvenile pikeperch brain.

Protamine-like proteins belong to sperm nuclear basic proteins and are related to somatic linker histone H1 family across the animal phylogeny (Eirín-López, et al., 2006) including fish (Saperas et al., 2006). Hence, this protein is so far exclusively known as sperm chromatin condensing protein and its function in the juvenile pikeperch brain remains currently unclear.

The family of matrilin proteins is thought to be involved in the formation of filamentous networks in the ECM of various tissues and interacts with collagen I (Piecha et al., 2002). The matrilin complex itself is a GO term belonging to ECM (The Gene Ontology Consortium, 2023). GO MF of matrilin is 'Enables $Ca^{2+}$ ion binding' (Burge et al., 2012). Matrilin-2 functions are known mostly in mammals (The UniProt Consortium, 2017). The *Mtn2* genes thus can be considered potentially linked to the ECM and collagen metabolism but also to ion transporters as numerous other transcripts consistently upregulated in piscivores.

The *krt13* gene-encoded keratin, type I cytoskeletal 13-like is a component of the intermediate filament network in epithelial cells (Hatta et al., 2018) with the only GO MF known – 'Enables structural molecule activity'. Additionally, in mice, this protein is known to be involved in protein metabolic and developmental processes, and in regulation of transcription in response to stress (Burge et al., 2012; Tang et al., 2019). This transcript should be further investigated in juvenile pikeperch neurotranscriptomics particularly in piscivores.

The *Mpk12* gene-encoded mitogen-activated protein kinase 12-like has the GO molecular functions 'enables ATP binding' and 'enables MAP kinase activity' and is involved in the GO biological processes 'intracellular signal transduction' and 'protein phosphorylation' (Burge et al., 2012; Tang et al., 2019).

The *Apoba* gene-encoded apolipoprotein Ba has the GO MFs 'Cholesterol transfer', 'Heparin binding', and 'Low-density lipoprotein particle binding', and its GO BPs involve 'Cholesterol homeostasis, metabolism and transport', 'Lipoprotein transport', and 'Triglyceride mobilization' (Burge et al., 2012; Tang et al., 2019). The heparin-binding function of this gene indicates its potential participation in the ECM of piscivores, while the cholesterol-related processes indicate a potential role of this transcript in synaptic plasticity, neuron development, cell signaling, and learning and memory (Hussain et al., 2019). Since this transcript showed high betweenness and centrality values it is supposed to substantially interact with other transcripts.

The *Gyc88E* gene-encoded soluble guanylate cyclase 88E-like is a protein with GO MF 'Enables guanylate cyclase activity' and 'Enables heme binding', the GO BP 'Involved_in cGMP biosynthesis, nitric oxide-cGMP-mediated signaling, and response to oxygen levels' (Burge et al., 2012; Tang et al., 2019). It participates in regulation of physiological functions including vasodilatation and neurotransmission (Derbyshire and Marletta, 2012). Its physiological functions relevant to CNS involve the

role of cGMP as modulator of neuronal excitability, synaptic transmission, and plasticity, contributing to learning and memory (Argyrousi et al., 2020). Hence, this transcript might be relevant to the juvenile pikeperch brain development.

The *Cltrn*-encoded collectrin (also known as Tmem27) is a transmembrane glycoprotein and an angiotensin-converting enzyme 2 homologue, known as a chaperone of amino acid transporters in the kidney and endothelium and from pancreas (Chu et al., 2023). This amino acid transport regulator might be functionally related with the numerous Slc transporters that were also upregulated in piscivores.

## Integration of transcription regulation and Slc transporters into the regulome

Slc transporters are directly linked with TFs that regulate their transcription, e.g. nuclear receptors, AP-1 (both transcribed in juvenile pikeperch brain as shown by this study) and other TFs; however, knowledge of these regulatory interactions is known mostly for mammals (e.g. Chan et al., 2013; Zhou and Shu, 2022) and far less for fish (Johnston and Kennedy, 2024). Slc transporters are further linked with the brain transcriptional machinery via metabolic pathways and changes in expression of (an)orexigenic neuropeptides – nutrient-sensing mechanisms regulating gene expression related to food intake and brain metabolism. Transcription of (an)orexigenic neuropeptides is regulated in response to fluctuating food availability (Conde-Sieira and Soengas, 2017) that accompanies both pikeperch phenotypes although in different ways. Planktivores are driven to filter zooplankton to grow as fast as possible. Piscivores have to utilize their structured habitat to hunt fish prey richer in nutrients than zooplankton and to hide from predators. Piscivores thus deliver more nutrients in their brain, whereby Slc transporters facilitate the uptake of nutrients into brain cells (Ayka and Şehirli, 2020). The intracellular nutrient levels are sensed by metabolic pathways, such as the mechanistic target of rapamycin (mTOR) and adenosine monophosphate-activated protein kinase (AMPK) (Soengas et al., 2018). These evolutionary conserved kinases act as master transcription regulators superior to TFs and drive cell metabolism, growth, and survival (Laplante and Sabatin, 2013; Lipton and Sahin, 2014). Their activity is driven by nutrient levels and the overall cellular energy status (Garza-Lombó et al., 2018). AMPK and mTOR pathways regulate cellular homeostasis at multiple levels, including the transcriptional regulation of genes linked to cell survival and bioenergetics. For example, AKT phosphorylation, which is influenced by mTORC2, regulates substrates like FOXO1/3a, a TF directly linking the metabolic pathways to gene transcription. This forms a feedback loop where Slc-mediated nutrient availability influences the metabolic state of brain cells, which then, through signaling pathways like AMPK/mTOR, modulates the activity of TFs. This represents a regulatory integration that allows the brain development and function to adapt dynamically to metabolic status and environmental nutrient availability during the distinct juvenile pikeperch phenotypes.

## Perspectives and conclusions

We link neurotranscriptomics of two juvenile pikeperch growth phenotypes with unprecedented details on their ecology, body size and their success in formation of the new-year class. Our results revealed that body size can be reflected in transcription of some genes (e.g. Slc transporters) and that transcription of other genes (e.g. TFs) can be rather related to the growth phenotype independently of the body size itself. Although based on the whole-brain transcriptome, which may have masked some details, our study represents an initial effort that showed the future directions

for investigating neurodevelopment in planktivorous and piscivorous pikeperch. We thus contribute to filling knowledge gaps in dynamics of early life history stages of non-model freshwater fishes as identified by Czeglédi et al. (2021). On the other hand, our study opened further questions about the specific role of the TFs upregulated in the juvenile pikeperch growth phenotypes and in the juvenile fish brain generally, i.e. whether activating or repressive and about their specific gene targets or the entire pathways. Further studies should focus on the moment of transition to piscivory, the key point from ecological and transcriptomics perspective. Finally, gene transcripts of more specific brain regions of the respective growth phenotypes will be the next steps towards understanding the neurodevelopmental drivers of juvenile pikeperch promoting this species to a novel model system.

## MATERIAL AND METHODS
### Ichthyological surveys and samples for RNA-seq and RNA extraction
Young-of-the-year pikeperch (*Sander lucioperca*) were sampled by two specialized ichthyological methods at Lipno Reservoir, Czech Republic in the last week of August 2022 and 2023. The young-of-the-year pikeperch individuals analyzed were too young and their sex was indistinguishable upon dissection since their gonads were not differentiated yet (Symonová et al., 2025). For the RNA-seq, four to five individuals of each phenotype and each year were sampled to ensure the lowest acceptable baseline of three replicates (Degen and Medo, 2025 and references therein) regarding the availability of the individuals of both phenotypes and trying to minimize our impact to the pikeperch population. The planktivorous individuals were caught by night pelagic trawling with a boat at the Lipno Reservoir as described by Jůza et al. (2014). Another round of pelagic trawling was performed in Mid-September 2022. The piscivorous individuals were caught by night littoral seining as described by Jůza et al. (2014). Size compositions of pikeperch in the last week of August of the second year of life were sampled by 40-50 CEN multimesh gillnets installed in all habitats of the reservoir according to Tesfaye et al. (2025). Fish age was established by otoliths reading as described by Tesfaye et al. (2023). Standard body size (SL; i.e. without tail) was recorded in the framework of the ichthyological survey for each fish each year (Table 1). The pikeperch individuals were killed by spinal cord interruption. The field sampling and experimental protocols used in this study were performed in accordance with the guidelines and permission from the Experimental Animal Welfare Commission under the Ministry of Environment of the Czech Republic (reference no. CZ 01679). The Experimental Animal Welfare Commission of Biology Centre of the Czech Academy of Sciences approved methods and ethics of the study. Their brains were immediately preserved with the RNA-Later solution (Thermo Fisher Scientific, USA). Brain RNA of each pikeperch individual was extracted using the RNA-Blue solution based on guanidinium thiocyanate-phenol-chloroform (Top-Bio, Czech Republic) according to the manufacturer's instructions. Obtained RNA was resuspended in RNAse-free ultrapure water (Top-Bio), quantified with Qubit™ 4 (Thermo Fisher Scientific) and Qubit™ RNA Broad Range Assay Kits (Invitrogen), and stored at −80°C. RNA samples of three individuals of each phenotype in each year (i.e. *n*=3 per phenotype and year totaling 12 individuals) with the best RNA yield and quality (RQN 8.6-10, Table S1) were sequenced by AZENTA (Germany) using the poly(A) enrichment.

### RNA-seq data filtering and read mapping
Sequence reads were trimmed to remove adapter sequences and nucleotides with poor quality using Trimmomatic v.0.36 (Bolger et al., 2014). The trimmed reads were mapped to the pikeperch reference genome SLUC_FBN_1.2 (GCA_008315115.1; Nguinkal et al., 2019) using the STAR aligner v.2.5.2b (Dobin et al., 2013). BAM files were generated. Sample sequencing statistics and statistics of mapping the reads to the reference genome of pikeperch is in Table S1.

Unique gene hit counts were calculated using featureCounts from the Subread package v.1.5.2 (Liao, et al., 2019). The hit counts were summarized and reported using the gene_id feature in the annotation file. Only unique reads that fell within exon regions were counted. The gene hit counts were normalized using DESeq2. A table of normalized gene hit counts was used for downstream differential expression analysis. Using DESeq2, a comparison of gene expression between the two phenotypes was performed intra-annually (i.e. in 2022 and 2023) and within phenotypes inter-annually (i.e. planktivores of 2022 against planktivores of 2023 and the same for piscivores). The Wald test was used to generate *P*-values and Log2FoldChanges. Genes with an adjusted *P*-value <0.05 and absolute Log2FoldChange >1 were considered as differentially expressed genes for each comparison (Love et al., 2014). A custom R script was used to assign a gene name to LOC identifiers of gene transcripts in the differential analysis table produced by AZENTA. In this way, 2583 gene names were obtained.

### Functional gene ontology, pathway enrichment and network centrality analyses
Pathway enrichment analyses and interaction statistics were performed online using the STRING database v.12 with the zebrafish (*Danio rerio*) gene annotation (Szklarczyk et al., 2023) and the tool Cytoscape (Shannon et al., 2003). Four sets of genes that resulted from intra- and inter-annual comparisons (Table 2) were uploaded in the database and analyzed for pathway enrichments. For these datasets, GO terms were explored, and local network clusters, KEGG (Kanehisa et al., 2017) and Reactome (Gillespie et al., 2022) pathways were identified. An Enrichment map plugin of Cytoscape was used to produce networks of the enriched pathways (Merico et al., 2010). Network centrality analysis was performed in Cytoscape, and the produced network centrality metrics (degree and betweenness) were used to identify potential hub genes *sensu* (Lehner et al., 2006; Rosati et al., 2024).

**Acknowledgements**
We thank Tomáš Tichopád for help with obtaining gene names of transcripts with LOC numbers and colleagues of the Fish Ecology Unit of the Institute of Hydrobiology, Biology Centre, Czech Academy of Sciences for help with sampling pikeperch.

**Competing interests**
The authors declare no competing or financial interests.

**Author contributions**
Conceptualization: R.S., J.K.; Data curation: R.S.; Formal analysis: R.S., M.B.; Funding acquisition: J.K.; Investigation: R.S., T.J., J.B.; Methodology: R.S., M.T., M.B., J.K.; Project administration: T.J.; Resources: T.J., M.T., M.B., Z.S., J.B., J.K.; Validation: R.S., M.T., J.K.; Visualization: R.S., Z.S.; Writing – original draft: R.S., J.K.; Writing – review & editing: R.S., T.J., M.T., M.B., Z.S., J.K.

**Funding**
This study has received funding from project No. R200962402 Fisheries management of a large reservoir under conditions of climate and nutrient change of the Regional cooperation programme of Czech Academy of Sciences and from the ELIXIR CZ Research Infrastructure (LM2023055, MEYS CR). This study was partially supported by the long-term strategic development financing of the Institute of Computer Science of Czech Academy of Sciences (RVO 67985807). Open Access funding provided by ELIXIR CZ LM2023055. Deposited in PMC for immediate release.

**Data and resource availability**
RNA-seq data for growth phenotypes and seasons are deposited in the Gene Expression Omnibus (GEO) under accession numbers GSE307083 and GSM9215499-GSM9215510. All other relevant data and details of resources can be found within the article and its supplementary information.

**Peer review history**
The peer review history is available online at https://journals.biologists.com/bio/lookup/doi/10.1242/bio.062280.reviewer-comments.pdf

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
