## [Peer Review File · Biology Open]

Differential activity of transcription factors and neuronal effectors during the development of pikeperch brain

Radka Symonová, Tomáš Jůza, Million Tesfaye, Marek Brabec, Zuzana Sajdlová, Jakub Brabec, Jan Kubečka and Radka Symonova

DOI: 10.1242/bio.062280

Editor: Lewis Halsey

Review timeline

Original submission: 23 September 2025

Editorial decision: 29 September 2025

First revision received: 13 October 2025

Accepted: 14 October 2025

Original submission

First decision letter

MS ID#: bio.062280

MS Title: Differential activity of transcription factors and neuronal effectors during the development of pikeperch brain

Authors: Radka Symonova; Radka Symonová; Tomáš Jůza; Million Tesfaye; Marek Brabec; Zuzana Sajdlová; Jakub Brabec; Jan Kubečka

Article Type: Research Article

I have now reached a decision on the above manuscript.

The reviewer reports are shown at the bottom of this email or can be accessed, together with a copy of this decision letter, by going to:

As you will see, the reviewers gave favourable reports, but raised some critical points that will require amendments to your manuscript. I hope that you will be able to carry these out, because we would like to be able to accept your paper.

At this stage, we also ask you to ensure your manuscript complies with our formatting guidelines - please see our manuscript preparation guidelines for details. Provided you are able to fully address the referees' comments, we are positive about publication of your paper (we accept over 95% of revision submissions) and therefore hope you won't mind any extra work involved in reformatting your manuscript at this point.

Please upload both a 'clean' version of your Word file, along with a highlighted version clearly showing where you have made changes in the revised manuscript. Please avoid using 'Track changes' in Word files as these are lost in PDF conversion.

I should be grateful if you would also provide a point-by-point response detailing how you have dealt with the points raised by the reviewers in the 'Response to Reviewers' box. Please attend to all of the reviewers' comments. If you do not agree with any of their criticisms or suggestions please explain clearly why this is so.

Reviewer 1

Overall comments:

In their study, Symonova et al attempt to link brain transcriptomic changes with seasonal and developmental phenotypes in the juvenile pikeperch, a species that is widespread across Europe. The authors surveyed fish populations in a specific field site, including determination of average body sizes, and then collected subsamples of animals for high-throughput brain transcript analysis. The resulting data showed strong partitioning of brain transcription factors and effector genes related to synaptic connections and general brain function. This partitioning was robust between different developmental stages. Overall, this paper provides a rare and robust evaluation of interactions between the environment development, and gene expression timing in the brain of a key fish species. The results are reasonably robust and will be interesting to a wide variety of researchers. The study is well-written, and the results are extensively discussed with robust referral to the literature. I am generally enthusiastic about this submission and have only a few relatively minor comments for the authors to consider.

Major comments:

1. This comment may reflect my ignorance of ichthyological surveys but how can the authors be sure which 2nd year populations derived from which 1st year populations (the arrows connecting 1st and 2nd year data in Fig. 1). How can the authors be sure that some of the larger animals from year 1 didn't then grow very slowly into year 2 and form part of the smaller sized cohort, and vice versa, how can the authors be sure that some of the smaller animals in year 1 didn't grow more rapidly and join the cohort of larger animals in year 2? Could such issues contribute to the lower similarities across years than between phenotypes in the same year (Fig. 2)?
2. Figure legends: Figures should be able to "stand alone" so please define all abbreviations used in each figure and figure legend independently. For example, in the legend for Fig. 2 please define MA, PCA, PC1. In table 2, define nc in NCRNA. Etc.
3. The discussion section of this paper is exhaustive but also a bit exhausting to read. At 9.5 pages, it is quite long and I recommend trimming this section by at least 1/3rd to improve the readability of your study.
4. Were any of the gene changes validated using PCR or other non-high throughput methods? It is fairly standard to select a few target genes that show robust changes and to then validate them using more time-consuming but accurate individual measures.

Minor Comments:

1. Consider including the species name in your title instead of "a juvenile fish"...specificity would improve communication here.
2. The manuscript is generally well-written but has several grammatical errors consistent with a non-native English writer. I recommend the authors seeks assistance in reviewing the grammar of their submission. Just a few examples:
 - o Line 67-68 - "...is driven by reaching a body size THAT ALLOWS a fish to swallow..."
 - o Line 70-72 - "Reaching the body size THAT enables A FISH to swallow..."
 - o Lines 89-81, etc.
3. This paper suffers from a bit of needless alphabet soup. Please consider using fewer abbreviations as their overuse makes the paper harder to follow. Uncommon abbreviations such as NEGs, IEGs, ECR, ECS, PPI, and YOYs are examples that could be written in full to improve readability. Other abbreviations that are only used 1x are likely not needed (e.g. ACE2). AA could be avoided too as it's only used a couple of times in the text. Etc. On the other hand, some abbreviations are used with definition. E.g. CNS on line 121 and in the discussion. On the other hand, lots of abbreviations are not defined in the results, e.g., SATB, ULD, TF Oxt, IGFBPs, VEGF,

VEGFR, etc. TF is defined a second time on line 309, in addition to being defined in the introduction. DT is defined in multiple places. Please be consistent.

4. Line 116: what is "a. o. "?
5. Line 152 - animals were killed, not euthanized. Please update.
6. Table 1 - what does the word "up" indicate in the last 2 columns?
7. Line 262 - check your decade
8. Do not abbreviate average as avg. in your results...it seems odd and hasn't been defined.
9. Second starting at line 380 is really a list of details without a lot of structure. This could maybe be moved to supplementary materials or presented as a table or figure instead of this long text description?

Matthew Pamerter

Reviewer 2

Manuscript Number: bio.062280

Manuscript Title: Differential activity of transcription factors and neuronal effectors during the development of a juvenile fish brain

Review Summary: This is a descriptive manuscript about the neural transcriptomic profiles of pikeperch fish that are either planktivores or piscivores. They describe how pikeperch fish go ontogenically from plantivorous diets to piscivorous (eating fish) diets in the wild. The data is based on wild caught fish from two different seasons (2022 and 2023), and their corresponding pikeperch phenotypes found on each season. Methodology is appropriate, and it did get me excited about expanding the neural toolbox for emerging models or non-model organisms, as these efforts will benefit the research community at large, in my opinion.

However, there are a few details that reduced enthusiasm. The major issue I found was the fact that neurotranscriptomics are too broad. Although I understand this part of the work is exploratory (and necessary before going deeper), transcription changes in the brain are difficult to disentangle or even make anything of them if analyzed in a big pool of entire/whole-brains. I strongly believe this work (or rather, maybe following work) should attempt to perform similar analysis in discrete regions (at least: telencephalon, optic tectum, etc...). If this is not possible for this manuscript, at least I suggest toning down conclusions, as this needs to be framed within what it is, which is a large pool of transcripts in which important changes might be or are definitely masked. Further, and I understand if not possible for this first manuscript, but having something functional would elevate it.

1. Experimental Quality

- a. Does each figure have the proper controls?

Yes

- b. Are experiments performed using appropriate methods that will answer the question (or test the hypothesis or support the observations) posed by the authors? Is the right tool used for the job?

Yes, but could be improved in resolution by focusing on discrete brain regions. Although, since these samples are collected from the wild, I understand the constraint here.

c. Were the data analyzed using appropriate statistical tests?

Yes.

2. Reproducibility

a. Were experiments in each figure performed using adequate number of biological replicates?

Yes, for transcriptomics and for wild caught.

b. Is there sufficient raw data to assess the rigor of the analysis?

There is, but it could benefit from a more discrete dissection of brain region, and increasing sample size. The fact that it depends on wild fish and seasons is an issue, but not one that can be easily resolved.

c. Does the methods section provide sufficient detail to permit reproducibility?

Yes.

3. Completeness

a. Are the author's conclusions supported by the data?

I feel some conclusions are a bit far-reaching in the sense that there are no functional experiments, but for the most part, yes.

b. Are there any flaws in the experimental design that invalidate the approach taken by the authors?

No.

c. Are there experiments that have not been performed, but if true would disprove the conclusion? If yes, and if such experiments would be costly or time-consuming to perform, do the authors acknowledge this in a discussion of the limitations?

The biggest one is the analysis of transcriptomics from the point of view of separate anatomical brain regions. I think this would only benefit there work, and would probably reveal even more genetic candidates at play, in relevant brain regions (relevant to the actual phenotype, and not just a mixture of transcripts that are changing concurrently with the phenotype and are not necessarily causative). I think this discussion and its limitation is lacking in the manuscript.

4. Scholarship

a. Do the authors cite and discuss the merits of relevant data that would argue against their conclusion?

Yes.

b. Do the authors cite and discuss the merits of relevant data that would support their conclusion?

Yes.

Line-by-Line Suggestions:

Here are a few minor corrections/suggestions, line by line:

88 - delete "in"

90 - revise grammar/syntax

132 - What do you mean by "significantly transcribed genes"? I suggest being more specific with differential gene expression vocabulary

186 - 188 - Revise syntax

220 - In Figure 1, body sizes are on x axis, not y axis. Number of individuals are on y axis.

Comment: is there a clearer way to show results in Figure 1? It is a bit confusing, especially with the diagonal arrows.

Suggestion: Re-arrange so in Panel A you have years 2022-2023 and in Panel B years 2023-2024.

Comment: What is "ordinary fingerlings?"

241 - 242 - Difficult to disentangle that reduction in planktivorous pikeperch was because death or transition to piscivory.

244 - What do you mean by "year class"? Which phenotype? Both?

262 - Correct "2033" to "2023". This also happens elsewhere.

312 - omit "a" in "a general cellular roles"

339 - 341 - revise for clarity

385 - 405 - suggest re-structuring as a figure or table, with complementary text.

Comment: I suggest improving how to combine the takehome message/narrative that the transition is correlated with burst/increase in transcription in brain. Also, which brain region? Unclear. At least maybe speculate. It cannot be the whole brain is controlling this transition.

427 - What do you mean by "lower brain activity"? I don't think there was any functional analysis here that measured brain activity.

444 - change "what is..." to "which is..."

Comment: How do you establish fish age? Unclear.

453 - 2033 again. Should be 2023.

509 - 510 - revise for clarity, add coma before "however" or separate sentences.

545 - don't use "a.k.a.", this sounds too conversational. Maybe just say "also known as...", the full phrase

582-589 - there are broad conclusions here regarding how Slc transporters impact transition from one phenotype to the other. This work does not have functional experiments that support this conclusion. Otherwise, at least comment/discuss potential future functional experiments that could answer this question. Highlight that most of these are speculative conclusions.

601 - coma after Faxdc2

Comment: The discussion could use a bit more synthesis. Right now, it reads like a list of "GO terms" that may be related instead of an organic discussion of findings and context.

652 - change "tough" for "through"? Actually not sure this is what authors meant to write, but if it is, revise.

Comment: In general, whole brain neuro-transcriptomics are too complex. It would be more informative to have these analyses from discrete brain regions and then you could associate function/speculate using better context. This is addressed in the closing sentences, but the discussion and manuscript throughout would benefit from highlighting this context more.

Reviewer's Responses to Questions

Experimental quality

Does each figure have the proper controls?

If 'No', please indicate reasons in Comments for Author box below.

Reviewer #1:

- Yes

Reviewer #2:

- Yes
-

Were the data analyzed using appropriate statistical tests?

If 'No', please indicate reasons in Comments for Author box below.

Reviewer #1:

- Yes

Reviewer #2:

- Yes
-

Reproducibility

Were experiments performed using adequate number of biological replicates?

If 'No', please indicate reasons in Comments for Author box below.

Reviewer #1:

- Yes

Reviewer #2:

- Yes
-

Does the methods section provide sufficient detail to permit reproducibility?

If 'No', please indicate reasons in Comments for Author box below.

Reviewer #1:

- Yes

Reviewer #2:

- Yes
-

Completeness

Are the manuscript's conclusions supported by the data?

If 'No', please indicate reasons in Comments for Author box below.

Reviewer #1:

- Yes

Reviewer #2:

- Yes

Scholarship

Do the authors cite and discuss the merits of data that would argue for and against their conclusion?

If 'No', please indicate reasons in Comments for Author box below.

Reviewer #1:

- Yes

Reviewer #2:

- Yes

Does the manuscript title & abstract accurately reflect the contents of the manuscript, without hyperbole?

If 'No', please indicate reasons in Comments for Author box below.

Reviewer #1:

- Yes

Reviewer #2:

- Yes

First revision

Author response to reviewers' comments

Dear Editor,

Thank you for reviewing our manuscript “Differential activity of transcription factors and neuronal effectors during the development of pikeperch brain”

Below, we provide our point-by-point response to Reviewers.

Reviewer 1

Dear Reviewer 1, thank you very much for your time and effort to review our manuscript and for your constructive feedback.

Major comments

1. This comment may reflect my ignorance of ichthyological surveys but how can the authors be sure which 2nd year populations derived from which 1st year populations (the arrows connecting 1st and 2nd year data in Fig. 1). How can the authors be sure that some of the larger animals from year 1 didn't then grow very slowly into year 2 and form part of the smaller sized cohort, and vice versa, how can the authors be sure that some of the smaller animals in year 1 didn't grow more rapidly and join the cohort of larger animals in year 2? Could such issues contribute to the lower similarities across years than between phenotypes in the same year (Fig. 2)?

Our response: This is in no way an ignorance but a very relevant comment. We do not have direct evidence what fraction of planktivores and piscivores from the first year of life contributes to what size groups in the second year of life. However, we have convincing indirect evidence based on evaluation of the contribution of either subcohorts to the 2nd year of life pikeperch for longer time

series (11 years) provided by Tesfaye et al. *in press*. A robust linear expression model revealed the proportions of 2nd year pikeperch recruited from either planktivorous and piscivorous phenotype. So, being sure about the proportions, it was possible to apportion the most likely origin of 2nd year pikeperch. We have corrected the legend to Figure 1 accordingly with the reference to the more detailed study of Tesfaye et al. *in press*.

2. *Figure legends: Figures should be able to "stand alone" so please define all abbreviations used in each figure and figure legend independently. For example, in the legend for Fig. 2 please define MA, PCA, PC1. In table 2, define nc in NCRNA. Etc.*

Our response: We have corrected the figure and table legends according to the Reviewer's comments, please see in red.

3. *The discussion section of this paper is exhaustive but also a bit exhausting to read. At 9.5 pages, it is quite long and I recommend trimming this section by at least 1/3rd to improve the readability of your study.*

Our response: We trimmed the section that was on page 9 in the previous version of our manuscript.

4. *Were any of the gene changes validated using PCR or other non-high throughput methods? It is fairly standard to select a few target genes that show robust changes and to then validate them using more time-consuming but accurate individual measures.*

Our response: At this stage, we have not performed any qPCR validation of our RNA-seq, partly because the GO terms and the overall patterns between the years were reproducible and partly because of the already long length of our manuscript.

Minor comments

1. We have included the species name "pikeperch" instead of the "a juvenile fish" as required by the Reviewer.

2. Thank you for your understanding of our non-native English. We have received feedback from a language check by a person with native English and corrected the examples given by the Reviewer.

3. We apologize for the "alphabet soup"! we wrote PPI, YOY and ACE2 in full and we wrote the abbreviations of Gene Ontology terms in full and only left the abbreviation in 2 cases when the next GO term contained the previous abbreviation in its name (insulin-like growth factor (IGF) transport and Uptake by IGF-binding proteins; Vascular Endothelial Growth Factor (VEGF) binding to VEGF receptors). We also removed the redundant definitions of TF and DT as indicated by Reviewer. The full name "differentially transcribed" was left in place only in the cases when not linked to word "genes". We would like to kindly ask Reviewer for agreeing with leaving the abbreviations ECR and ECS - ECS occurs 13 times and ECR 6 times.

4. We have changed "a. o." to "among others"

5. We changed "euthanized" by "killed" and we apologize for this inaccuracy.

6. The legend of the Table 1 already contains explanation of "up" as "significantly upregulated genes".

7. We have corrected "2033" to "2023" and thank you very much for noticing this issue!

8. We have removed the abbreviations of avg. and wrote it in full as required by Reviewer.

9. We have reorganized the paragraph to fit the manuscript, and we apologize for the list-like form of this section.

Reviewer 2

Dear Reviewer 2, thank you very much for your time and for your effort to review our manuscript and for your constructive feedback.

However, there are a few details that reduced enthusiasm. The major issue I found was the fact that neurotranscriptomics are too broad. Although I understand this part of the work is exploratory (and necessary before going deeper), transcription changes in the brain are difficult to disentangle or even make anything of them if analyzed in a big pool of entire/whole-brains. I strongly believe this work (or rather, maybe following work) should attempt to perform similar analysis in discrete regions (at least: telencephalon, optic tectum, etc...). If this is not possible for this manuscript, at least I suggest toning down conclusions, as this needs to be framed within what it is, which is a large pool of transcripts in which important changes might be or are definitely masked. Further, and I understand if not possible for this first manuscript, but having something functional would elevate it.

Our response: we understand this issue, and we were trying to separate diencephalon, however, the brain is very small particularly in planktivores. This was the reason why we performed a microCT investigation of the brain to be able to delimit the relevant brain regions. Unfortunately, this work is still in progress, and we could not apply knowledge obtained from microCT here. We highly appreciate Reviewer 2 for his/her understanding of this situation, and we will do our best to follow these instructions in our future work. As required, we toned down conclusions on lines 665-667: Although based on the whole-brain transcriptome, which may have masked some details, our study represents an initial effort that showed the future directions for investigating neurodevelopment in planktivorous and piscivorous pikeperch.

Minor corrections

Line 88 - we have deleted the redundant “in”

Line 90 - we have revised grammar to improve clarity

Line 132 - we have added the missing “differentially” and thank you for noticing this omission.

Lines 186-188 - we have revised syntax as required by the Reviewer.

Line 220 - we have corrected the axes names and thank the Reviewer for noticing this mistake! We have changed the explanation inside the graph from “Ordinary fingerlings” to “Planktivores”, which is the correct wording.

Lines 241-242 - we have explained how this reduction in planktivores was evidenced by our ichthyological surveys in 2022 and 2023

Line 244 - we defined “year class” and explained that in the next year(s) of life all are piscivorous.

Line 262 + line 453 - we have corrected the incorrect 2033 to 2023 on both places in the manuscript and thank the Reviewer for indicating this mistake!

Line 312 - we have deleted “a” on this line and thank the Reviewer for indicating this mistake!

Line 339-341 - We have split the long sentence and reordered it to increase clarity as required by the Reviewer.

Line 385-405 - We have reorganized the paragraph to fit the manuscript, and we apologize for the list-like form of this section.

Comment: I suggest improving how to combine the takehome message/narrative that the transition is correlated with burst/increase in transcription in brain. Also, which brain region? Unclear. At least maybe speculate. It cannot be the whole brain is controlling this transition.

Our response: we have added a sentence “Nonetheless, it remains to be identified, what specific brain region is responsible for the increase of gene transcription.”

Line 427 - We have changed the incorrect “lower brain activity” to “gene transcription”, which is more suitable for the context.

Line 444 - We changed “what is” to “which is”

Comment on how do you establish fish age?

Our response: we have established fish age by otoliths reading, which we have added into the Material and Methods section and apologize for this omission.

Line 509-510 - We have revised and separated sentences to increase clarity as.

Line 545 - We have removed “a.k.a.” and replaced by “also known as” everywhere as.

Line 582-589 We have rephrased the part to express the uncertainty of conclusions as required by Reviewer 2. We added another literature reference to support our ideas and added a concluding sentence “However, these expectations need to be confirmed by functional experiments.”.

Line 601 - we corrected the sentence.

Comment: The discussion could use a bit more synthesis. Right now, it reads like a list of “GO terms” that may be related instead of an organic discussion of findings and context.

Our response: Here, we wanted to avoid speculation without having evidence of any functions. Where our results enabled the synthesis was involved, please, check the green parts of the section.

Line 652 - we have changed “tough” for “though”

Comment: In general, whole brain neuro-transcriptomics are too complex. It would be more informative to have these analyses from discrete brain regions and then you could associate function/speculate using better context. This is addressed in the closing sentences, but the discussion and manuscript throughout would benefit from highlighting this context more.

Our response: We have addressed this issue at the beginning of our response to Reviewer 2. Please, check our very first response.

On behalf of all coauthors & With kind regards
Radka Symonova

Second decision letter

MS ID#: bio.062280R1

MS TITLE: Differential activity of transcription factors and neuronal effectors during the development of pikeperch brain

AUTHORS: Radka Symonova; Radka Symonová; Tomáš Jůza; Million Tesfaye; Marek Brabec; Zuzana Sajdlová; Jakub Brabec; Jan Kubečka

I have read through your rebuttal, and the associated changes to your manuscript, and I am happy to tell you that your manuscript has now been accepted for publication in Biology Open, pending our standard publication integrity checks. It was accepted on 14th October 2025.